# Co-Existing Vestibular Hypofunction Impairs Postural Control, but Not Frailty and Well-Being, in Older Adults with Benign Paroxysmal Positional Vertigo

**DOI:** 10.3390/jcm14082666

**Published:** 2025-04-14

**Authors:** Sara Pauwels, Nele Lemkens, Winde Lemmens, Kenneth Meijer, Pieter Meyns, Raymond van de Berg, Joke Spildooren

**Affiliations:** 1Faculty of Rehabilitation Sciences, REVAL-Rehabilitation Research Centre, Hasselt University, 3590 Diepenbeek, Belgium; joke.spildooren@uhasselt.be; 2Department of Otorhinolaryngology and Head and Neck Surgery, School for Mental Health and Neuroscience, Faculty of Health Medicine and Life Sciences, Maastricht University Medical Centre, 6229 Maastricht, The Netherlands; raymond.vande.berg@mumc.nl; 3Department of Otorhinolaryngology, Head and Neck Surgery ZOL Hospital, 3600 Genk, Belgium; nele.lemkens@kno.be (N.L.); winde.lemmens@kno.be (W.L.); 4Department of Nutrition and Movement Sciences, NUTRIM Institute of Nutrition and Translational Research in Metabolism, Maastricht University, 6229 Maastricht, The Netherlands; kenneth.meijer@maastrichtuniversity.nl

**Keywords:** older adults, benign paroxysmal positional vertigo, vestibular hypofunction, repositioning maneuver, postural control, frailty, well-being, caloric irrigation test

## Abstract

**Background:** Vestibular hypofunction occurs in 29.5% of older adults with benign paroxysmal positional vertigo (BPPV), but its impact on postural control, well-being and frailty was not studied before. This study compared the well-being, frailty and postural control between older adults with BPPV and vestibular hypofunction (oaBPPV+), and older adults with only BPPV (oaBPPV). **Methods:** Thirty-one older adults (≥65 years old) diagnosed with BPPV were recruited. Unilateral vestibular hypofunction was defined as a >25% caloric asymmetry, and bilateral vestibular hypofunction as a total response <6°/s per ear, using bithermal caloric irrigations. The oaBPPV+ group was compared to the oaBPPV group using the measures of well-being (Dizziness Handicap Inventory, Falls Efficacy Scale and 15-item Geriatric Depression Scale), frailty (Modified Fried Criteria), and postural control (timed chair stand test, mini-Balance Evaluation Systems test and Clinical Test of Sensory Interaction on Balance (CTSIB)). Falls and the number of repositioning maneuvers were documented. Significance level was set at α = 0.05. **Results:** Unilateral vestibular hypofunction was present in 32% of participants, mainly in females (*p* = 0.04). Bilateral vestibular hypofunction was not found. The oaBPPV+ group (*n* = 10, mean age 72.5 (4.5)) experienced more comorbidities (*p* = 0.02) than the oaBPPV group (*n* = 21, mean age 72.6 (4.9)). Groups did not differ regarding dizziness symptoms (*p* = 0.46), fear of falling (*p* = 0.44), depression (*p* = 0.48), falls (*p* = 0.08) or frailty (*p* = 0.36). However, the oaBPPV+ group showed significantly worse postural control under vestibular-dependent conditions (*p* < 0.001). **Conclusions:** Despite equally impaired well-being and frailty, the oaBPPV+ group showed greater sensory orientation deficits. Clinicians and researchers should be alert for co-existing vestibular hypofunction in older adults with BPPV, since this may exacerbate their already impaired postural control more than only BPPV.

## 1. Introduction

Dizziness, vertigo and unsteadiness are common symptoms among older adults. Benign paroxysmal positional vertigo (BPPV) is diagnosed in 39.1% of the older adults with vertigo [1] and is most likely caused by otoconia migrating from the utricle into the semicircular canals. In 85–90% of cases, the posterior semicircular canal is affected, due to its anatomical position [2]. Approximately 50% to 70% of the BPPV cases are considered to be idiopathic. The remaining cases of BPPV are secondary and associated with underlying conditions such as head trauma, vascular issues or co-existing vestibular disorders. [3]

Attacks of nystagmus and vertigo when the head is moved in the plane of the canals, are typically reported symptoms [4]. However, older adults with BPPV frequently report general dizziness, lightheadedness and unsteadiness between attacks [5,6]. They also have a significantly increased fall incidence and number of falls, decreased cognitive performance and an increased level of frailty compared to their peers. In addition to physical symptoms, older adults with BPPV also experience reduced well-being, with an increased fear of falling and feelings of depression [7,8,9].

The gold-standard treatment with repositioning maneuvers (RMs) [10] (aiming to relocate the dislodged otoconia) are less effective in older adults. More RMs are needed to resolve BPPV [11]. Older adults with BPPV have decreased improvement in postural control compared to younger adults with BPPV [7] and an increased risk of recurrence and residual dizziness [12] after treatment with RM. It remains controversial whether older age is an independent risk factor for both residual dizziness and recurrence, possibly due to the increased fragmentation and demineralization of otoconia, or if the elevated risk is attributable to a higher prevalence of other risk factors in older adults, such as delayed diagnosis, reduced mobility, impaired vestibular compensation and greater comorbidity or prevalence of co-existing vestibular disorders [12,13].

Co-existing vestibular hypofunction is commonly observed in patients with BPPV [14], affecting 29.5% of older adults specifically [15]. Vestibular hypofunction implies a loss of function in one or both vestibular organs. This can be related to the semicircular canals or the otolith organs [16]. In this study, vestibular hypofunction was defined as a reduced response on the caloric irrigation test [17]. It can lead to a wide variety of symptoms such as visually induced dizziness, cognitive complaints, tiredness, moderate-to-severe dizziness related handicap and impairments in multiple domains of postural control [18,19]. Vestibular hypofunction can have different etiologies such as trauma, an underlying vestibular disorder (e.g., Ménière’s disease, vestibular neuritis) or extensive damage of the vestibular end-organ [20].

In adults with BPPV, co-existing vestibular hypofunction does not affect the prognosis on the resolution of nystagmus after RM, but increases the risk of residual dizziness [21] and recurrence [22]. These previous studies always included both younger and older adults and did not investigate the impact on daily living. However, older adults with BPPV and co-existing vestibular hypofunction (oaBPPV+) may experience a prolonged duration of symptoms.

Therefore, this study aimed to compare the well-being (dizziness-related handicap, fear of falling and feelings of depression), frailty, postural control and a number of RMs needed for the resolution of nystagmus, between older adults with BPPV and vestibular hypofunction (oaBPPV+), and older adults with BPPV alone (oaBPPV). We hypothesized that the oaBPPV+ group would be more frail and would have more severely impaired postural control and well-being compared to older adults with only BPPV (oaBPPV). 

## 2. Materials and Methods

This study was approved by the ethical committees of Hospital Oost-Limburg, Genk (ZOL Genk) and Hasselt University (B3712021000013, date of approval 31 May 2021) and complied with the Declaration of Helsinki. This study was registered at ClinicalTrials.gov (NCT03526653).

### 2.1. Participants

As part of a larger prospective study [7,23], community-dwelling older adults (≥65 years old) diagnosed with BPPV were recruited at the Department of Otorhinolaryngology at ZOL Genk, between September 2021 and July 2023. Older adults were screened for the eligibility criteria and invited to participate by the first author (S.P.). The inclusion criteria were as follows: (1) a diagnosis of posterior semicircular canal BPPV or lateral semicircular canal BPPV (geotropic or apogeotropic variant) [4] without prior treatment for the current episode, (2) the ability to stand independently for at least 30 s and (3) the ability to walk at least 10 m, with or without walking aids. 

Exclusion criteria included: (1) inability to understand and follow simple instructions, (2) temporary or permanent residence in a psychiatric or residential care center, hospital or rehabilitation facility, (3) contraindications for diagnostic maneuvers or caloric irrigation testing, (4) the presence of a neurodegenerative disorder, (5) being in the rehabilitation phase following an orthopedic or cardiovascular event, (6) prior treatment for the current episode of BPPV and (7) the resolution of BPPV before pre-treatment data collection was completed. Patients in the oaBPPV group with previously diagnosed peripheral vestibular disorders or central vestibular disorders were not excluded from this study.

### 2.2. Study Design

At T0, informed consent was obtained and a BPPV diagnosis was made using video-Frenzel goggles (VisualEyes™ 505 Video Frenzel system Interacoustics, Middelfart, Denmark) (Appendix A). Demographic data (age, sex, weight, height, use of walking aid, sleeping pattern, comorbidities, number of medications) were collected. Comorbidities were inquired with a checklist presented in Appendix A. The duration of the symptoms was inquired and classified as “some days”, “several weeks” or “several months”. To screen for cognitive impairment, the Montreal Cognitive Assessment (MOCA) [24] was used at T1 (maximum seven days after recruitment).

#### 2.2.1. Well-Being

The Dutch versions of the Dizziness Handicap Inventory (DHI) [25], Falls Efficacy Scale (FES-I) [26] and 15-item Geriatric Depression Scale (GDS-15) [27] were used to assess the dizziness-related handicap, fear of falling and feelings of depression. Questionnaires were filled out at home and checked for completeness at T1. 

The DHI is a 25-item questionnaire designed to assess the impact of dizziness on daily activities, comprising physical (7 items), emotional (9 items) and functional (9 items) subscales. Scores range from 0 (best) to 100 (worst) [25]. The Dutch version has excellent test–retest reliability and is proven highly reliable to assess the dizziness-related handicap [28].

The FES-I is a 16-item questionnaire that measures an individual’s concern about falling using a 4-point Likert scale, ranging from 1 (not concerned) to 4 (very concerned) [26]. It demonstrates excellent test–retest reliability and validity in individuals with vestibular disorders [29]. The Dutch version has acceptable reliability and construct validity [28].

The GDS-15 is a sensitive (86%) and specific (79%) yes/no questionnaire used to screen for depression in older adults, with scores ranging from 0 (no depression) to 15 (severe depression) [30].

#### 2.2.2. Frailty

Frailty was assessed using the Fried criteria, including unintentional weight loss, self-reported exhaustion, slowness, weakness and self-reported physical inactivity [31], as modified by Avila-Funes et al. [32]. This adjustment was made to ensure feasibility within the broader protocol. Participants were classified as ‘‘frail’’ if they had three or more frailty components among the five criteria, they were considered ‘‘prefrail’’ if they fulfilled one or two frailty criteria, and ‘‘robust’’ if none. 

Fall history and the number of falls over the past 12 months were questioned. The reasons for falls were categorized as dizziness-related, accidental, syncope or unknown.

#### 2.2.3. Postural Control

Postural control was objectively measured with inertial sensors (APDM Wearable Technologies). These sensors were bilaterally secured on the wrist and feet, as well as the sternum and the fifth lumbar vertebrae. The following tests and outcome parameters were used:During the timed chair stand test [33], the participant was asked to stand up from a chair five times as fast as possible with the arms held against the chest. Total time (s), sit-to-stand time (s) and stand-to-sit time (s) were derived from the sensors on the sternum and lumbar vertebrae. An increased time implies poorer performance.The mini-balance evaluation systems test (Mini-BESTest) [34] was used to calculate the total score and subscores for anticipatory postural control, reactive postural control, sensory orientation and dynamic gait. A decreased score implies poorer performance.Some sub-items were evaluated in detail:○For the timed up and go with and without a dual task (TUG and TUG_dualtask_), the total time (s), sit-to-stand time (s), stand-to-sit time (s) and turn duration (s) were derived from the sensors on the sternum and lumbar vertebrae. An increased time implies poorer performance. The dual task cost was calculated as $\frac{\left(dual\;task\;time-single\;task\;time\right)}{single\;task\;time}\times 100$ [35]. A greater absolute value for the dual task cost (DTC) implicates poor performance deterioration under the dual task condition.○For the 10-m walk test (10 MWT) at preferred gait speed and with head turns (10 MWHT), the gait speed (m/s), cadence (steps/min), stride length (m), stride length standard deviation (SD), double support (% gait cycle time), gait cycle duration (s) and gait cycle duration SD were derived from the mean of the bilateral sensors on the feet. A decreased gait speed, cadence, stride length with an increased stride length SD, double support, gait cycle duration and gait cycle duration SD implies poorer performance○For the longest trial of the worst side of the unilateral stance, the total time (s), sway area (m^2^/s^4^), mean velocity (m/s), path length (m/s^2^) and range (m/s^2^) of accelerations were derived from the lumbar sensor. A decreased time and increased sway area, mean velocity, path length and range of acceleration implies poorer performance.

The Clinical Test of Sensory Interaction on Balance (CTSIB) [36] was used to evaluate the relative contribution of sensory systems to postural control during six standing conditions: (1) firm surface with eyes open (CTSIB1), (2) firm surface with visual dome (CTSIB2), (3) firm surface with eyes closed (CTSIB3), (4) foam surface with eyes open (CTSIB4), (5) foam surface with visual dome (CTSIB5) and (6) foam surface with eyes closed (CTSIB6). Data collected from the lumbar sensor included the total time (s), sway area (m^2^/s^4^), mean velocity (m/s), path length (m/s^2^) and range (m/s^2^) of accelerations. A decreased time and increased sway area, mean velocity, path length and range of acceleration implies poorer performance. All assessments for postural control were conducted without the use of a walking aid.

#### 2.2.4. Treatment with Repositioning Maneuvers

After the assessment for well-being, frailty and postural control, the presence of BPPV was re-confirmed with video-Frenzel, and treatment with repositioning maneuvers (RM) was performed. Canalolithiasis of the posterior canal was treated with the Epley maneuver. Cupulolithiasis of the posterior canal and less mobile participants with posterior canal BPPV were treated with the Semont maneuver. Lateral semicircular canal BPPV was treated with the Gufoni and modified Gufoni maneuvers for the geotropic and apogeotropic variant, respectively [10].

After the first RM, the presence of nystagmus during a diagnostic maneuver was re-assessed with video-Frenzel goggles, and a second RM was performed if necessary (and if tolerated by the patient). This was repeated weekly, until no nystagmus was observed during diagnostic maneuvers. The number of RMs needed until the resolution of nystagmus during a diagnostic maneuver was recorded. A subdivision of the number of RMs needed until the resolution of nystagmus, according to the BPPV diagnosis, was analyzed descriptively.

#### 2.2.5. Caloric Irrigation Test

Approximately one month after the resolution of nystagmus (T2), a bithermal caloric irrigation test was carried out by a trained examiner, using warm (44 °C) and cold (30 °C) water flow of 250 mL/30 s (Aqua Stim™ caloric irrigator, Interacoustics, Middelfart, Denmark) alternating between the right and left ear. The first author was blinded to the results of the caloric irrigation tests when possible (i.e., *n* = 20), until all participants completed the study. Eye movements were recorded with videonystagmography and the maximum slow phase eye velocity was analyzed with VisualEyes™, or visual inspection in the case of unreliable values. The percentage of canal paresis or hypofunction was calculated using the Jongkees Index formula. In line with the terminology of the Barany diagnostic criteria for bilateral vestibulopathy [37], the term hypofunction will be used throughout this paper. For unilateral vestibular hypofunction, the cut-off value was set at 25%, with an absolute value of the healthy side within the range of 20°/s to 83°/s [20,38]. A sum of both responses per ear <6°/s was defined as bilateral vestibular hypofunction [37]. Older adults with BPPV and co-existing unilateral or bilateral vestibular hypofunction according to the above described criteria were categorized oaBPPV+. Older adults with BPPV and a caloric irrigation test result within normal ranges were categorized as oaBPPV.

### 2.3. Statistics

Data analysis was performed using IBM SPSS statistics software (version 25.0 for Windows, SPSS Inc., Armonk, NY, USA). GraphPad Prism 10 (GraphPad Software, San Diego, CA, USA) was used to create graphs.

As this is a secondary analysis of a larger prospective study, and data on oaBPPV+ patients are lacking in the literature, no a priori sample size calculation could be performed. Therefore, a sensitivity power analysis for the current sample size at 80% power and α= 0.05 was performed, using G*Power (Version 3.1.9.6). A Cohen’s d of 0.97 was required to achieve 80% power at α = 0.05 for well-being, mini-BESTest, unilateral stance and timed chair stand-test which were analyzed with the Mann–Whitney U test (oaBPPV+ *n* = 10 vs. oaBPPV *n* = 21). A Cohen’s d of 0.56 was required to achieve 80% power at α = 0.05 when frailty was analyzed with the Chi-square test (oaBPPV+ *n* = 10 vs. oaBPPV *n* = 21). Effect sizes were classified as small (d = 0.2), medium (d = 0.5) or large (d ≥ 0.8) [39]. Normality of the data was assessed using Shapiro–Wilk tests. Significant outliers were detected using Tukey’s method [40] and excluded if deemed necessary by consensus. Continuous data were analyzed with unpaired *t*-tests and Mann–Whitney U tests for normally and non-normally distributed data, respectively. Categorical data and frailty were analyzed with the Pearson Chi-square test. Effect sizes for non-parametric tests were calculated as Cohen’s d according to Fritz et al. [39]. Normally distributed data are expressed as the mean (standard deviation), non-normally distributed data as the median (interquartile range). To analyze a group-, test-, and group χ test interaction effect for CTSIB, TUG and TUG_dualtask_, and for the 10-m walk test (10 MWT) and 10-m walk test with head turns (10 MWHT), linear mixed models were fit for each outcome to compensate for possible random missing values [41]. Main effects were reported as f-values and *p*-values.

A Bonferroni correction was used for post-hoc comparisons within each mixed model. Alpha values of the questionnaires, frailty, postural control and physical activity were adjusted to correct for multiple comparisons using the Holm–Bonferroni sequentially rejective procedure [42]. The Holm–Bonferroni correction was applied within the following groups: well-being (DHI and subscales, FES-I and GDS-15), frailty (total score and subscores), mini-BESTest (total score and subscores) and each test for postural control separately.

## 3. Results

Unilateral vestibular hypofunction was found in 10 (32.2%) out of the 31 included older adults with BPPV (Appendix A). No bilateral vestibular hypofunction was found. There was no significant difference in age (*p* = 0.77), weight (*p* = 0.25) and height (*p* = 0.16) between the oaBPPV+ group (*n* = 10, 9 females, mean age 72.5 (4.5)) and the oaBPPV group (*n* = 21, 10 females, mean age 72.62 (4.86)). However, the oaBPPV+ group included significantly more females (*p* = 0.04) and had a higher prevalence of comorbidities (*p* = 0.03) compared to the oaBPPV group (Table 1). Specifically, hypercholesterolemia was reported significantly more often in the oaBPPV+ group (*p* = 0.02), while other comorbidities did not differ significantly between groups (Table 2).

In eight out of ten oaBPPV+ patients, the vestibular hypofunction was at the ipsilateral side of the BPPV diagnosis. Two other oaBPPV+ patients, both with posterior semicircular canal BPPV, had vestibular hypofunction at the contralateral side. Two out of ten oaBPPV+ patients with lateral semicircular canal BPPV had vestibular hypofunction at the ipsilateral side of the BPPV diagnosis. All walking aids had been in use prior to the onset of BPPV. There was no significant difference in BPPV diagnosis or the number of RMs needed for the resolution of nystagmus. The number of RMs needed until the resolution of nystagmus according to BPPV diagnosis is presented in Appendix A.

### 3.1. Well-Being

The dizziness-related handicap (total score and subscores), fear of falling and feelings of depression did not significantly differ between the oaBPPV+ and oaBPPV groups, as measured by the DHI (oaBPPV+: 33.6 (14.6), oaBPPV: 34.6 (17.6), *p* = 0.46), FES-I (oaBPPV+: 28.2 (11.3), oaBPPV: 27.2 (10.3), *p* = 0.44) and GDS-15 (oaBPPV+: 3.5 (3.25), oaBPPV: 3 (3.5), *p* = 0.48) (Figure 1 and Appendix A).

### 3.2. Frailty

Frailty status (*p* = 0.36) nor unintentional weight loss (*p* = 0.35), self-reported exhaustion (*p* = 0.45), slowness (*p* = 0.22), weakness (*p* = 0.56) or physical inactivity (*p* = 0.69), significantly differed between the oaBPPV+ and oaBPPV groups (Table 3).

### 3.3. Falls

Four out of ten (40%) oaBPPV+ participants and seven out of twenty-one (33.3%) oaBPPV participants experienced a fall in the past twelve months. The fall incidence (*p* = 0.5), odds of falling (OR 1.3; 95% CI [0.28,6.3]; *p* = 0.08), number of falls (*p* = 0.4) and reason for falls were not significantly different between groups. Three oaBPPV+ participants reported their falls as accidental, while one oaBPPV+ participant fell due to dizziness. Four oaBPPV participants reported their falls as accidental, while three oaBPPV participants reported dizziness as the cause of their falls (Table 3).

### 3.4. Postural Control

There was a trend towards a decreased total score (20.5 (8.7) vs. 22 (5), *p* = 0.02) of the mini-BESTest in the oaBPPV+ group compared to the oaBPPV group. The sensory orientation subscore was significantly decreased in oaBPPV+ (4.5 (3)) compared to oaBPPV (6 (0), *p* = 0.001). There was a trend towards a decreased anticipatory postural control subscore (3.5 (1.5) vs. 5 (1.5), *p* = 0.02) and dynamic gait subscore (8 (3) vs. 8 (2), *p* = 0.04) in the oaBPPV+ group. Cohen’s d, however, was too small (<0.97) according to the sensitivity analysis, indicating less than 80% power. The reactive postural control subscore did not significantly differ between groups (oaBPPV+: 5 (2.8), oaBPPV: 5 (3), *p* = 0.33) (Figure 2).

The total time (*p* = 0.32), sit-to-stand time (*p* = 0.48) and stand-to-sit time (*p* = 0.2) of the timed chair stand test were not significantly different between groups.

The oaBPPV+ group had a significantly shorter duration for the unilateral stance compared to the oaBPPV group (*p* = 0.01). The sway area (*p* = 0.29), velocity (*p* = 0.21), path (*p* = 0.22) and range (*p* = 0.47) did not differ (Table 3).

For the CTSIB (Figure 3, Appendix A), a significant group X condition interaction effect was found for the sway area (F_5,174_ = 5.54, *p* < 0.001), path length (F_5,174_ = 5.74, *p* < 0.001), range (F_5,174_ = 4.65, *p* < 0.001) and time (F_5,174_ = 6.56, *p* < 0.001).

Post-hoc comparison revealed that the oaBPPV+ group had a significantly larger sway area and range, and a significantly shorter performance time during CTSIB5 and CTSIB6 compared to the oaBPPV group. Within the oaBPPV+ group, the sway area and range were significantly larger, and performance time was significantly shorter during CTSIB5 and CTSIB6 compared to CTSIB1, CTSIB2, CTSIB3 and CTSIB4. Within the oaBPPV group, however, the range of CTSIB6 was only significantly larger compared to the range in CTSIB1. Sway area and time did not differ between the conditions in the oaBPPV group.

Post-hoc comparison revealed that the oaBPPV+ group had a significantly longer path length during CTSIB5 compared to the oaBPPV group. Within the oaBPPV+ group, the path length during CTSIB5 was significantly longer than during CTSIB1, CTSIB2, CTSIB3, CTSIB4 and CTSIB6. Within the oaBPPV group, the path length did not differ between the conditions.

For the sway velocity, no significant group X condition interaction effects or group effects were found. Both groups (F_5,174_ = 6.7 *p* < 0.001) had an increased sway velocity in CTSIB6 compared to CTSIB1, CTSIB2, CTSIB3, CTSIB4 and CTSIB5.

For the TUG and TUG_dualtask_ (Table 4), no significant group X condition interaction effects or group effects were found. Both groups needed significantly more time (F_1,58_ = 8.4, *p* = 0.005) to complete the TUG_dualtask_ compared to the TUG. The sit-to-stand time, stand-to-sit time and turn duration did not significantly differ between the conditions.

The dual task cost did not significantly differ (*p* = 0.19) between the oaBPPV+ (30.3 (−0.43–70.2)) and oaBPPV (20.2 (0.18–128.6) groups.

For the 10 MWT and 10 MWHT (Table 4), no significant group X condition interaction effects were found.

There was a trend towards a slower gait speed (F_1,58_ = 5.1, *p* = 0.03), a slower cadence (F_1,58_ = 6.83, *p* = 0.01) and shorter stride length (F_1,58_ = 5.03, *p* = 0.03) during both conditions in the oaBPPV+ group compared to the oaBPPV group. Both groups had a significantly slower gait speed (F_1,58_ = 9.8, *p* = 0.003), shorter stride length (F_1,58_ = 8.9, *p* = 0.004) and increased double support time (F_1,58_ = 13.8, *p* ≤ 0.001) during the 10 MWHT compared to the 10 MWT. There was a trend towards a slower cadence (F_1,58_ = 4.3, *p* = 0.04) and longer cycle duration (F_1,58_ = 5.2, *p* = 0.03) in both groups during the 10 MWHT compared to the 10 MWT.

## 4. Discussion

The present study aimed to compare the postural control, falls, frailty and well-being between the oaBPPV+ and oaBPPV groups. Thirty-two percent of the included older adults with BPPV had co-existing unilateral vestibular hypofunction. Significantly more women had vestibular hypofunction. The oaBPPV+ group reported significantly more comorbidities and had significantly more pronounced impairments in sensory orientation, particularly in vestibular-dependent conditions compared to the oaBPPV group. The oaBPPV+ group also had a decreased performance time for the unilateral stance compared to the oaBPPV group and walked slower. In contrast to our hypothesis, well-being and frailty status did not differ between both groups. The presence of vestibular hypofunction had no effect on the number of RMs needed to resolve nystagmus. Previous research indicated that older adults with BPPV have increased odds of falling [43], decreased well-being and postural control, and are more (pre)frail compared to older adults without BPPV [7]. Clinicians and future researchers should therefore be alert for a co-existing vestibular hypofunction, since this may exacerbate their already impaired postural control, more than BPPV alone.

A prevalence of 32.3% of vestibular hypofunction in older adults with BPPV is in agreement with the existing literature and more specifically Song et al. [15]. In their study, a prevalence of 29.5% in older adults was reported, although the cut-off for an abnormal caloric test was not reported. Albeit small, the sample of the oaBPPV+ group included significantly more women and had more comorbidities than the oaBPPV group. It is known that women are more affected by vestibular disorders, possibly due to hormonal differences [44,45]. 

An older onset age of BPPV, the female sex, comorbidities and co-existing vestibular hypofunction are all independently reported risk factors for residual dizziness and recurrence in patients with BPPV [12,22,46,47]. While further research is necessary, including assessments of residual dizziness and recurrence, our findings suggest a potential interrelationship between these risk factors. They might indicate the presence of co-existing vestibular hypofunction in older adults with BPPV.

The oaBPPV+ group demonstrated significantly increased postural sway and shorter performance time in CTSIB5 and CTIB6 compared to the oaBPPV group. As these are vestibular-dependent tasks, it can be assumed that these results are caused by the co-existing vestibular hypofunction and not the significant difference in sex or comorbidities. The oaBPPV+ group experienced more difficulties to sensory reweight the input from the vestibular system, when visual and somatosensory input is altered, to maintain their postural control. Our findings support the previous work of Cohen et al. [48], suggesting that CTSIB can serve as a part of a screening battery for vestibular impairments in older adults. Previous research, which excluded patients with BPPV and a co-existing vestibular disorder confirmed through vestibular function tests, also indicated that sensory orientation and spatiotemporal parameters is impaired in older adults with BPPV alone [49,50,51]. Yet our results suggest that co-existing vestibular hypofunction can exacerbate these deficits. Although improvements in CTSIB after RMs have been reported in older adults with BPPV, future studies should investigate whether similar gains are also observed in the oaBPPV+ group, or if additional rehabilitation is necessary.

The lack of significant differences in CTSIB1-4 and gait assessments (10 MW(H)T and TUG) suggests that oaBPPV+ can compensate for their unilateral vestibular hypofunction when visual and somatosensory input is available. However, there was a trend towards slower gait speed, slower cadence, and shorter stride length during walking in the oaBPPV+ group compared to the oaBPPV group. The lack of significant differences in gait assessment may also be due to the small sample size. Accordingly, there was no significant difference in the incidence, odds of falling, or number of falls between the oaBPPV+ and oaBPPV groups. Nevertheless, one third of all participants reported at least one fall. Moreover, the increased odds of falling in patients with BPPV, decreased gait speed and positive treatment effect of RMs, was highlighted in previous research [43]. It is therefore recommended to screen, and treat, BPPV in older adults with impaired postural control to prevent possible falls [10].

In contrast to our hypotheses, no difference in well-being and frailty was found between the oaBPPV+ and oaBPPV groups. We hypothesized that the oaBPPV+ group would be more frail and would have more severely impaired well-being compared to the oaBPPV group, due to their prolonged duration of symptoms. However, no difference in the duration of complaints was observed between groups, as the majority of the oaBPPV group also experienced their symptoms for several months. Previous research already indicated that older adults with BPPV can experience a prolonged time to diagnose due to their atypical symptoms, or the false belief of patients and clinicians that dizziness is a normal part of aging [52]. Nevertheless, in both groups, more than half of the participants are (pre-)frail, the average DHI score indicated a moderate dizziness-related handicap, and there was a moderate to high concern for falls. Since these factors did not differ between the groups, they may be attributed to the presence of BPPV. However, further research including a control group without vestibular disorders or with UVH only is needed to confirm this hypothesis.

Our findings encourage the use of objective vestibular function tests to screen for co-existing vestibular disorders when including older adults with BPPV in future research. After all, older adults with BPPV and co-existing unilateral vestibular hypofunction may not report significantly greater dizziness-related handicaps or frailty, but their postural control could be more severely affected. In a previous systematic review on postural control in patients with BPPV, only nine out of thirty-seven included studies used vestibular functions tests to screen for co-existing vestibular disorders [53]. The other studies may therefore have included participants with BPPV and a co-existing vestibular disorder, resulting in greater impaired postural control than would be present in BPPV alone.

This study has some limitations. As the secondary analysis of a larger prospective study, no a priori power calculation was conducted, resulting in small and unequally distributed groups. In 11 participants, the researcher who administered the tests and treatments also performed the caloric test. This researcher was therefore not blinded to all results. There was no control group of older adults without BPPV included. At the moment, the post-treatment assessment of postural control, frailty and well-being is not yet available. This limits the ability to isolate the specific effects of BPPV or co-existing vestibular hypofunction.

This study only used the caloric test to assess vestibular function, without incorporating other complementary vestibular tests like Video Head Impulse Testing. This limited the assessment of vestibular hypofunction, as the caloric test predominantly stimulated the horizontal semicircular canal and the superior vestibular nerve at a low frequency. Nevertheless, if performed correctly, the caloric test seems to be most sensitive for detecting vestibular hypofunction [16]. The assessments for postural control and the caloric irrigation test were not performed immediately upon diagnosis, leading to exclusion of a substantial number of participants due to the resolution of their BPPV before data collection was completed.

Finally, due to the elaborate study protocol and recruitment through outpatient care, participants were required to delay treatment and return to the hospital for multiple visits; this may have favored the inclusion of more mobile and less impaired participants. Accordingly, the inclusion criteria with minimal requirements for postural control may have favored more mobile participants. Consequently, the study sample may not fully represent the broader population of older adults with BPPV, and the differences between the oaBPPV and oaBPPV+ groups may have been reduced.

## 5. Conclusions

This is the first partially blinded study comparing postural control, frailty and well-being between the oaBPPV+ and oaBPPV groups. Despite a similar well-being and frailty status as oaBPPV, oaBPPV+ experienced more pronounced impairments in sensory orientation, particularly in vestibular-dependent conditions. Clinicians and future researchers should therefore be alert for co-existing vestibular hypofunction in older adults with BPPV, since this can exacerbate their already impaired postural control more than BPPV alone.

## Figures and Tables

**Figure 1 jcm-14-02666-f001:**
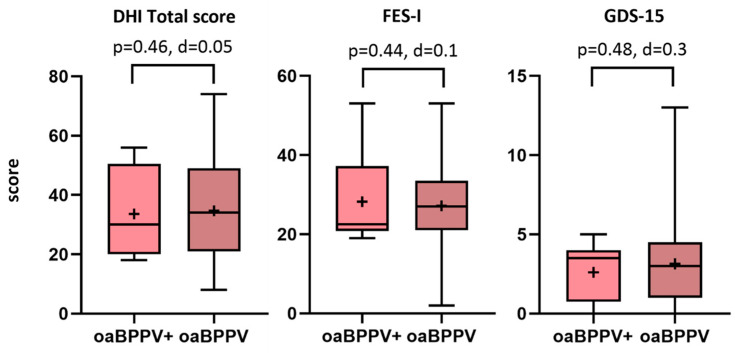
DHI, FES-I and GDS-15 scores in older adults with BPPV (*n* = 21) and co-existing vestibular hypofunction compared to older adults with BPPV (*n* = 10). The boxplots display the medians, interquartile range, and minimum and maximum values, with the ‘+’ symbol representing the mean values. Abbreviations: oaBPPV+, older adults with BPPV and vestibular hypofunction; oaBPPV, older adults with BPPV; DHI, dizziness handicap inventory; FES-I, falls efficacy scale international; GDS-15, 15-item geriatric depression scale.

**Figure 2 jcm-14-02666-f002:**
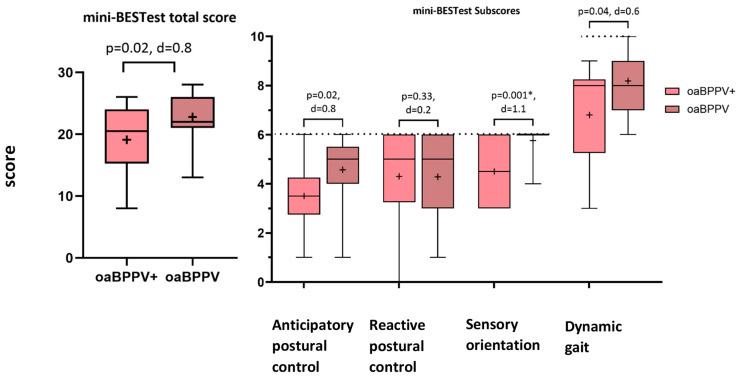
Mini-BESTest total scores and subscores in older adults with BPPV (*n* = 21) and co-existing vestibular hypofunction compared to older adults with BPPV (*n* = 10). The boxplots display the median, interquartile range, and minimum and maximum values, with the ‘+’ symbol representing the mean values. Dotted lines indicate the maximum score possible on subscales. Significant *p*-values after Holm–Bonferroni correction are indicated with ‘*’. Abbreviations: mini-BESTest; mini Balance Evaluation Systems test; oaBPPV+, older adults with BPPV and vestibular hypofunction; oaBPPV, older adults with BPPV.

**Figure 3 jcm-14-02666-f003:**
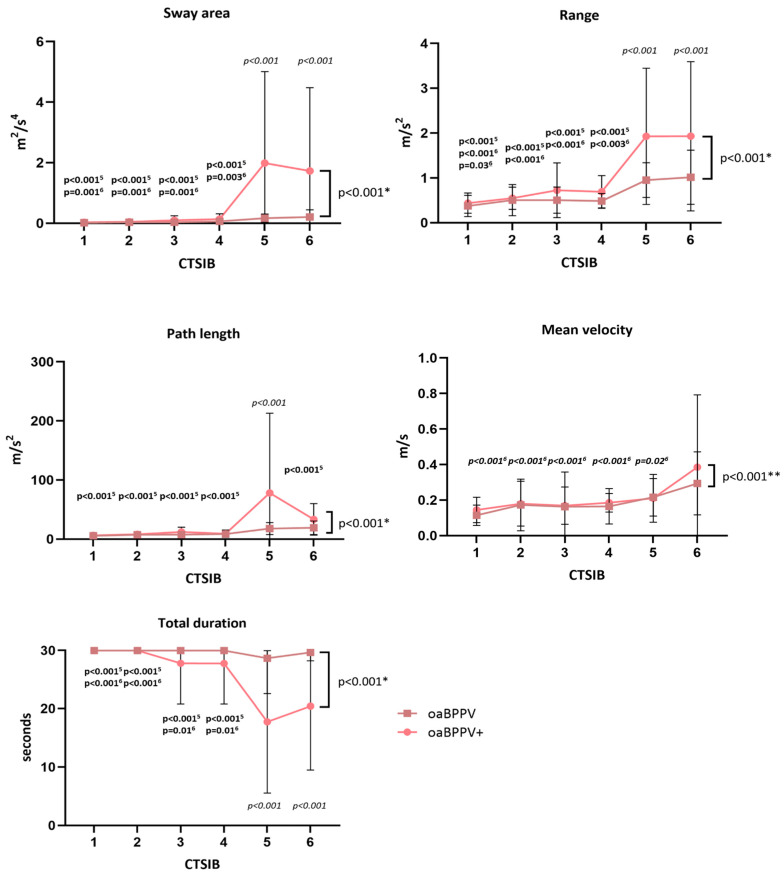
Results (means and standard deviations) of the Clinical Test of Sensory Interaction on Balance in older adults with BPPV (*n* = 21) and co-existing canal paresis compared to older adults with BPPV (*n* = 10). ‘*’ indicates a significant group X condition interaction effect; ‘**’ indicates a significant condition effect. *p*-values in italics indicate significant post-hoc comparisons between groups after significant group X condition interaction effects. *p*-values in bold indicate significant post-hoc comparisons within groups after significant group X condition interaction effects. *p*-values in bold and italics indicate significant post-hoc comparisons after condition effects. ‘^5^’ indicates a significant difference with CTSIB5 from the oaBPPV+ group. ‘^6^’ indicates a significant difference with CTSIB6 from the oaBPPV+ group. ‘^6^’ indicates a significant difference with CTSIB6 from the oaBPPV group. Abbreviations: CTSIB, Clinical Test of Sensory Interaction on Balance; CTSIB1, standing on a firm surface with eyes open; CTSIB2, standing on a firm surface with a visual dome; CTSIB3, standing on a firm surface with eyes closed; CTSIB4, standing on a foam surface with eyes open; CTSIB5, standing on foam surface with a visual dome; CTSIB6, standing on a foam surface with eyes closed; oaBPPV+, older adults with BPPV and vestibular hypofunction; oaBPPV, older adults with BPPV.

**Table 1 jcm-14-02666-t001:** Subject characteristics.

Characteristics	oaBPPV+ *n =* 10	oaBPPV *n =* 21	*p*-Value
Female/Male	9/1	10/11	0.04
Age	72.5 (4.5)	72.62 (4.86)	0.95
Weight (kg)	77.35 (11.45)	75.99 (11.12)	0.88
Height (m)	1.64 (0.06)	1.66 (0.09)	0.54
BPPV			0.25
RPSCC (*n*, %)	4, 40%	9, 43%	
LPSCC (*n*, %)	4, 40%	9, 43%	
Bilateral PSCC (*n*, %)	0, 0%	0, 0%	
RLSCC geotropic (*n*, %)/apogeotropic (*n*, *%*)	1, 10%/0, 0%	0, 0%/3, 14%	
LLSCC geotropic (*n*, %)/apogeotropic (*n*, %)	1, 10%/0, 0%	0, 0%	
Number of repositioning maneuvers	2 (3)	2 (1.75)	0.2
Caloric irrigation test			
UVH ipsilateral BPPV (*n*, %)	8, 80%	0, 0%	
UVH contralateral BPPV (*n*, %)	2, 20%	0, 0%	
No vestibular hypofunction (*n*, %)	0, 0%	21, 100%	
Duration of complaints			0.1
Some days (*n*)	0, 0%	3, 14%	
Several weeks (*n*)	3, 30%	2, 10%	
Several months (*n*)	7, 70%	16, 76%	
Walking aid			0.12
None (*n*)	7, 70%	20, 95%	
Crutch (*n*)	2, 20%	1, 5%	
Walker (*n*)	1, 10%	0, 0%	
Sleeping pattern			0.56
Good (*n*)	7, 70%	11, 52%	
Restless (*n*)	1, 10%	7, 33%	
Long time needed to fall asleep (*n*)	1, 10%	1, 5%	
Restless + long time needed (*n*)	1, 10%	2, 10%	
Number of medications	5.7 (2.41)	4.33 (2.72)	0.09
MOCA total score	23.2 (4.85)	24 (3.45)	0.3

Significance level was set at α = 0.05. Significant differences according to the unpaired *t*-test are indicated in bold. Normally distributed data are expressed as mean (SD), non-normally distributed data as median (interquartile range). Abbreviations: oaBPPV+, older adults with BPPV and vestibular hypofunction; oaBPPV, older adults with BPPV; BPPV, benign paroxysmal positional vertigo; RPSCC, right posterior semicircular canal BPPV; LPSCC, left posterior semicircular canal BPPV, RLSCC, right lateral semicircular canal BPPV; LLSCC, left lateral semicircular canal BPPV; UVH, unilateral vestibular hypofunction; MOCA, Montreal Cognitive Assessment scale.

**Table 2 jcm-14-02666-t002:** Comorbidities.

Comorbidities	oaBPPV+	oaBPPV	*p*-Value
Number of comorbidities	4 (2)	2 (3)	0.03
Cardiovascular (*n*, %)	2, 20%	6, 29%	0.2
Cerebrovascular (*n*, %)	1, 10%	1, 5%	0.2
Diabetes mellitus (*n*, %)	1, 10%	4, 19%	0.2
Hypertension (*n*, %)	8, 80%	11, 52%	0.07
Hypercholesterolemia (*n*, %)	9, 90%	11, 53%	0.02
Vitamin D deficiency (*n*, %)	3, 30%	5, 24%	0.25
Osteoporosis (*n*, %)	3, 30%	3, 15%	0.14
Other (*n*, %)	6, 60%	9, 43%	0.3

Significance level was set at α = 0.05. Significant differences according to the Chi-square test are indicated in bold. Abbreviations: oaBPPV+, older adults with BPPV and vestibular hypofunction; oaBPPV, older adults with BPPV.

**Table 3 jcm-14-02666-t003:** Frailty.

Frailty	oaBPPV+	oaBPPV	*p*-Value	Cohen’s d
Robust (*n*, %)	3, 30%	5, 25%	0.36	0.3
Pre-frail (*n*, %)	3, 30%	9, 45%		
Frail (*n*, %)	4, 40%	6, 30%		
Unintentional weight loss				
Yes (*n*, %)/No (*n*)	1, 10%/9	5, 25%/15	0.35	0.3
Self-reported exhaustion				0.2
Yes (*n*, %)/No (*n*)	6, 60%/4	10, 50%/10	0.45	
Slowness				0.5
Yes (*n*, %)/No (*n*)	5, 50%/5	6, 30%/14	0.22	
Weakness				0.1
Yes (*n*, %)/No (*n*)	3, 30%/7	7, 35%/13	0.56	
Physical inactivity				0
Yes (*n*, %)/No (*n*)	8, 80%/2	4, 20%/16	0.69	
Timed chair stand test				
Total time (s)	17.1 (11.6)	16.2 (6.14)	0.32	0.18
Sit-to-stand time (s)	1 (0.3)	1 (0.4)	0.48	0.02
Stand-to-sit time (s)	0.8 (0.3)	0.7 (0.2)	0.2	0.39
Unilateral stance				
Area (m^2^/s^4^)	1.4 (2.6)	1.1 (2.1)	0.29	0.19
Velocity (m/s)	0.3 (0.4)	0.4 (0.4)	0.21	0.31
Path (m/s^2^)	73.7 (47.14)	59.1 (60.7)	0.22	0.28
Range (m/s^2^)	2.2 (2.8)	2.4 (3)	0.47	0.03
Time (s)	4.8 (8.9)	13.4 (14.3)	0.01	0.84
Falls				
Fall history Yes (*n*, %)/No (*n*)	4, 40%/6	7, 33%/14	0.5 OR 1.3; 95% CI [0.28,6.3]; 0.08	0.1
Number of falls				
0 (*n*, %)	6, 60%	14, 67%	0.4	0.3
1 (*n*, %)	2, 20%	3, 14%		
2 (*n*, %)	2, 20%	3, 14%		
>2 (*n*, %)	0, 0%	1, 5%		
Reason for falls				
Accidental (*n*, %)	3, 30%	4, 19%	0.4	0.3
Dizziness (*n*, %)	1, 10%	3, 14%		
Syncope (*n*, %)	0, 0%	0, 0%		
No falls (*n*, %)	6, 60%	14, 67%		

Significant differences according to Mann–Whitney U test after Holm–Bonferroni corrections are indicated in bold. Normally distributed data are expressed as mean (SD), non-normally distributed data as median (interquartile range). Abbreviations: oaBPPV+, older adults with BPPV and vestibular hypofunction; oaBPPV, older adults with BPPV; OR, odds ratio; CI, confidence interval.

**Table 4 jcm-14-02666-t004:** Results of the timed up and go, timed up and go with dual task, 10-m walk test and 10-m walk test with head turns.

Timed Up and Go	oaBPPV+	oaBPPV
Total time (s)	13.8 (3.8)	12.2 (3.1)
Sit-to-stand time (s)	1.1 (0.3)	1.1 (0.5)
Stand-to-sit time (s)	0.9 (0.2)	0.9 (0.2)
Turn time (s)	2.4 (0.4)	2.4 (0.6)
Timed up and go with dual task		
Total time (s)	16.6 (8.1)	13.4 (5.9)
Sit-to-stand time (s)	1.1 (0.3)	1 (0.3)
Stand-to-sit time (s)	0.9 (0.2)	0.9 (0.3)
Turn time (s)	2.7 (0.6)	2.7 (0.6)
Dual task cost (%)	30.4 (21.5)	26.4 (30.1)
10-m walk test		
Gait speed (m/s)	0.9 (0.2)	1 (0.2)
Cadance (step/min)	99.9 (16)	107.2 (10)
Stride length (m)	1 (0.17)	1.1 (0.2)
Stride length SD (m)	0.04 (0.03)	0.3 (0.02)
Double support time (%GCT)	22.4 (4.8)	21.8 (2.8)
Cycle duration (s)	1.2 (1.3)	1.1 (0.1)
Cycle duration SD (s)	0.03 (0.01)	0.03 (0.2)
10-m walk test with head turns		
Gait speed (m/s)	0.7 (0.22)	0.8 (0.2)
Gait speed (m/s)	0.9 (0.2)	1 (0.2)
Cadance (step/min)	99.9 (16)	107.2 (10)
Stride length (m)	1 (0.17)	1.1 (0.2)
Stride length SD (m)	0.04 (0.07)	0.3 (0.05)
Double support time (%GCT)	22.4 (4.8)	21.8 (2.8)
Cycle duration (s)	1.2 (1.3)	1.1 (0.1)

Normally distributed data are expressed as mean (SD), non-normally distributed data as median (interquartile range). Abbreviations: oaBPPV+, older adults with BPPV and vestibular hypofunction; oaBPPV, older adults with BPPV.

## Data Availability

The data presented in this study are available on request from the corresponding author due to the nature of data (patient information).

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
