# Peer review of "Co-Existing Vestibular Hypofunction Impairs Postural Control, but Not Frailty and Well-Being, in Older Adults with Benign Paroxysmal Positional Vertigo"

_jcm, 2025, doi:10.3390/jcm14082666_

Round 1
Reviewer 1 Report
Comments and Suggestions for Authors
I would like to thank you the JCM for the opportunity to review this interesting manuscript.
Even if some of my concerns were properly adressed in the limitations part of the Discussion, I still have some concerns:
1)Even if the authors demonstrated a worse postural control in oaBPPV+ patients, the same result may be expected when comparing patients who do not suffer from bppv with or without unilateral vestibular hypofunction (in other words, the difference in terms of postural control may be expected independently from BPPV, so what's new?). Please discuss this point.
2)In table 1 few cases of horizontal semicircular canals (HSC) BPPV were reportated. What about vestibular hypofunction in this cases? Was the HSC canal involved by otolith displacement also affected by hypofunction? Please discuss this point.
Reviewer 2 Report
Comments and Suggestions for Authors
I would like to start by thanking you for the opportunity to review this manuscript. This investigation focuses on the co-occurrence of vestibular hypofunction and BPPV in elderly patients. The topic is both interesting and has the potential to contribute to current knowledge. Treating BPPV in older patients presents challenges in clinical practice. There are several issues that need to be addressed.
Keywords
The keywords should specify that caloric testing was used to assess vestibular functioning.
Introduction
Lines 45-46. In this context, ’dizziness’ is misleading, as BPPV typically causes vertigo.
Lines 48-49. The phrase ‘General dizziness, lightheadedness and unsteadiness between attacks are..’ is a fragment and should be rephrased for clarity.
Considering the general introduction of BPPV, it should be highlighted that BPPV primarily affects the posterior semicircular canals, as these are the parts of the inner ear most influenced by gravity. Additionally, it is important to note that secondary cases of BPPV can arise, such as those associated with Ménière’s disease. Regarding this.
Lines 53-57. Providing explanations for the higher risk of BPPV recurrences and residual symptoms in older patients would be beneficial.
Line 66. Please correct the misspelling of Ménière’s disease.
It is important to emphasise the significant correlation between dizziness, handicap, and psychiatric symptoms regarding the impact of BPPV symptoms on daily life and overall well-being.
Lines 82-83. Please ensure that you also include the date of ethical approval.
Considering the diagnosis of BPPV, the most recent ICVD guideline should be referenced:
doi: 10.3233/VES-150553.
Considering exclusion criteria, it should be clarified whether previously diagnosed peripheral vestibular disorders or central vestibular disorders were considered as exclusion criteria.
Line 104. In medical writing, it is important to consistently use the term ’sex’ rather than ’gender.’
It would be more appropriate to include separate subsections for each measurement used in this study. For example, the questionnaires, especially the DHI, should be presented in greater detail in a dedicated subsection. This section should specifically address the subscores of the DHI that were utilised in the study's results. Additionally, it is important to include a statement regarding whether the DHI has been validated in the primary language of the study population.
Lines 165-166. Providing an average number of necessary manoeuvres for symptom-free periods would be beneficial.
It is important to clarify that caloric testing specifically measures the functionality of the horizontal semicircular canal and the superior vestibular nerve in response to low-frequency stimuli. Additionally, when assessing unilateral hypofunction, the primary parameter that can be calculated is canal paresis. Regarding this, please cite the following reference article:
doi: 10.1080/14992027.2022.2059711
Line 196. When presenting median values, it is more appropriate to include the IQRs along with the Q1 and Q3 values. Additionally, presenting the ranges of Cohen’s d values would be beneficial.
Results
Lines 212-213. The methods should specify the groups (oaBPPV+ and oaBPPV) and their abbreviations. Additionally, the methods lack an analysis of comorbidities and a statement on which comorbidities were considered in this investigation.
Table 1. The abbreviation ‘N’ for females and males is ambiguous. Furthermore, the names of the statistical tests and the significance level should be detailed in the table caption. Additionally, ‘positional’ is the correct term for BPPV instead of ‘positioning’.
Table 2. The table caption should provide a more detailed description. Additionally, a more accurate format for presenting the results would be ’yes/no (%)’, as the current format may confuse readers. Furthermore, the term ‘diabetes mellitus’ would be more appropriate. The methods section should clarify whether both type 1 and type 2 diabetes were considered, and the same clarification is needed for hypertension.
Line 238. It is confusing that non-significant differences in the DHI subscores are mentioned, as only the total scores are presented in Figure 1.
Figure 1. The figure legend should specify the significance level.
Table 3. The table caption should offer a more detailed description. It would also be beneficial to include the ranges for interpreting Cohen's d. Additionally, the caption should specify the name of the statistical test used and the significance level. Furthermore, the abbreviations ’CI’ and ’OR’ need to be defined for clarity.
Lines 260-262. It would be helpful to include the specific p-values in the text.
Line 270. Please provide the specific p-value for this significantly decreasing tendency.
Line 275. Please place Figure 2. before its explanation in the text.
Line 280-311. In light of this subsection, it would be beneficial to not only collect data but also to provide practical explanations for readers.
Figure 2 and Figure 3 should specify their significance levels in the captions.
Discussion
Lines 344-345. The abbreviations for the two groups should be defined upon their first use, and thereafter, only the abbreviations should be utilised.
Lines 346-347. Providing potential explanations for this female predominance would be of interest.
Lines 350-351. It would be beneficial to attempt to explain the non-significant results.
Line 363. Please correct ‘gender’ to ‘sex’.
Explanations for sex differences should be provided, including the potential effects of hormonal fluctuations in women, particularly during menopause.
Lines 391-392. A more accurate statement would be that older patients should be screened for BPPV and treated to prevent falls.
Line 414. It should be highlighted that caloric testing is limited to analysing the function of the horizontal canals and the superior vestibular nerves.
Conclusions
Lines 432-433. The conclusion should not reference previous investigations but focus on the findings of the current study.
I am looking forward to receiving the revised version of the manuscript, which includes a point-by-point response to each review comment with all required changes accurately made. This is necessary for deciding whether this manuscript can be considered.
Round 2
Reviewer 2 Report
Comments and Suggestions for Authors
Thank you for sending the revised version of the manuscript. The authors have made significant efforts to improve its quality, and it reads much better now. However, some of the previous reviewer comments were not fully addressed. Please ensure that each review comment is considered before this manuscript can be considered for publication. Additionally, I would like to point out that Figure 2 is not referenced in the text; this should be done for all figures.
Author Response
Response to Reviewer 2 Minor revision comments
2. Quality of English Language
( ) The English could be improved to more clearly express the research.
(x) The English is fine and does not require any improvement.
3. Questions for General Evaluation
|
Yes |
Can be improved |
Must be improved |
Not applicable |
|
|
Does the introduction provide sufficient background and include all relevant references? |
( ) |
( ) |
(x) |
( ) |
|
Is the research design appropriate? |
( ) |
(x ) |
() |
( ) |
|
Are the methods adequately described? |
( ) |
( ) |
(x) |
( ) |
|
Are the results clearly presented? |
( ) |
(x) |
( ) |
( ) |
|
Are the conclusions supported by the results? |
(x) |
() |
( ) |
( ) |
- Point-by-point response to Comments and Suggestions for Authors
Thank you for sending the revised version of the manuscript. The authors have made significant efforts to improve its quality, and it reads much better now. However, some of the previous reviewer comments were not fully addressed. Please ensure that each review comment is considered before this manuscript can be considered for publication. Additionally, I would like to point out that Figure 2 is not referenced in the text; this should be done for all figures.
Response: Thank you for giving me the opportunity to submit a revised draft of my manuscript “A co-existing vestibular hypofunction impairs postural control, but not frailty and well-being, in older adults with Benign Paroxysmal Positional Vertigo” to Journal of Clinical Medicine. In the major revision, we’ve provided a point-by-point response in which each comment was addressed, and adjustments were made in the paper where deemed necessary. Given the nonspecific comment and the fact that only the methods section is marked as ‘must be improved’, we have made the following revisions, considering the reviewer’s comment from the major revision:
- In response to comment 14, we have added canal paresis as a outcome parameter of the caloric irrigation test, but that we will use the term ‘hypofunction’ throughout the paper. This terminology is in line with the terminology of the Barany diagnostic criteria for bilateral vestibulopathy (doi: 10.3233/VES-17061 ), the consensus document for vestibular hypofunction (doi: 10.1007/s00415-020-10139-4) and recent research on chronic vestibular hypofunction (doi:3390/jcm13185381).
This was revised as ‘The percentage of canal paresis or hypofunction was calculated using the Jongkees Index formula. In line with the terminology of the Barany diagnostic criteria for bilateral vestibulopathy, the term hypofunction will be used throughout this paper.’ The revision can be found at line 247-249. - Figure 2 was referenced in text at line 376, and marked in yellow. All other figures and tables were also referenced in text and indicated in yellow.